# Pain-stimulated ultrasound vocalizations and their impact on pain response in mice

Satoka Kasai[1]*, Saki Ukai[1], Junpei Kuroda[1], Tsugumi Yamauchi[2], Daisuke Yamada[2], Akiyoshi Saitoh[2], Satoshi Iriyama[3], Masashi Suzuki[4], Kazuki Arita[5], Yoshio Nakano[5], Satoru Miyazaki[5], Kazumi Yoshizawa[1]

1 Laboratory of Pharmacology and Therapeutics, Department of Pharmacy, Faculty of Pharmaceutical Sciences, Tokyo University of Science, Katsushika-ku, Tokyo, Japan, 2 Laboratory of Pharmacology, Department of Pharmacy, Faculty of Pharmaceutical Sciences, Tokyo University of Science, Katsushika-ku, Tokyo, Japan, 3 Laboratory of Quantum information dynamics, Department of Information Sciences, Faculty of Science and Technology, Tokyo University of Science, Noda, Chiba, Japan, 4 FUJIMIC, Inc., Tokyo, Japan, 5 Laboratory of Bioinformatics, Department of Pharmacy, Faculty of Pharmaceutical Sciences, Tokyo University of Science, Katsushika-ku, Tokyo, Japan

* satoka_kasai@rs.tus.ac.jp

## Abstract

Pain is a complex phenomenon encompassing both the physiological and psychological aspects of sensation and emotion, respectively. In recent years, pain has been clarified to arise even without direct injury, with emotional transmission as a cause. However, the specific mechanisms behind emotional transmission are still not well understood. In this study, sounds in the ultrasonic domain that were recorded during pain stimulation in mice were used as sound stress to examine the effects of psychological stress caused by exposure to ultrasound on tactile thresholds. We also examined the effects of psychological stress caused by the ultrasound on an inflammatory pain model of mice. The tactile threshold decreased the next and three days after sound stress exposure in mice. DNA microarray analysis of the mouse thalamus exposed to sound stress revealed increased expression of inflammation-related genes, *Prostaglandin-endoperoxide synthase 2* and *C-X-C motif chemokine ligand 1*. Their respective inhibitors, loxoprofen and SB225002, significantly improved hyperalgesia induced by sound stress. When sound stress was applied to a mouse model of inflammatory pain in which pain thresholds were restored 14 days after complete Freund's adjuvant administration, prolonged pain and attenuated analgesic effects of loxoprofen were observed. These results suggest that sound stress not only induces inflammation in the brain that causes hyperalgesia but may also be partially responsible for exacerbating inflammatory pain, hence complicating treatment.

**Data availability statement:** All DNA microarray analysis data files are available from the NCBI Gene Expression Omnibus (accession number: GSE272934). Other relevant data are within the paper and its Supporting information.

**Funding:** SK, DY, AS, SI, YN, SM, and KY have received a grant from FUJIMIC Inc. (Tokyo, Japan, https://www.fujimic.com). The sponsors do not have any role in this manuscript submission. SK, DY, AS, and KY have received a grant from AMED-CREST (Grant Number JP24gm1510008, https://www.amed.go.jp/koubo/16/02/1602C_00011.html). The funders do not have any role in this manuscript submission.

**Competing interests:** I have read the journal's policy and the authors of this manuscript have the following competing interests: [This work was supported by a grant from FUJIMIC Inc. (Tokyo, Japan) and AMED-CREST under Grant Number JP24gm1510008. MS is an employee of FUJIMIC Inc. This does not alter our adherence to PLOS ONE policies on sharing data and materials.]

## Introduction

Pain is an important physiological phenomenon that can be a major stressor, sometimes to the point of death, and can occur even in the absence of actual tissue damage [1]. Many reports show that psychological stress enhances pain [2,3]. In 2016, Smith *et al.* demonstrated that "bystander" mice housed in the same environment as mice undergoing inflammatory pain exhibited similar hyperalgesia [4]. This emotional transmission is thought to occur because of various information (sight, hearing, smell, etc.) obtained from others; however, their mechanisms are unclear.

Ultrasound is defined as sound inaudible to humans, above 20 kHz [5]. Mice emit various ultrasonic vocalizations in different social contexts at both audible and ultrasonic frequencies [6,7]. Rodents produce a wide variety of ultrasonic vocalizations in various social situations. However, emotional transmission through ultrasound vocalizations remains unexplored. Therefore, we conducted a study on the pain vocalized by squeaks, particularly in the ultrasound domain to show that the ultrasound vocalization of mice under pain affected hyperalgesia in other mice.

Here, we provide reports of the pain response in mice exposed to the ultrasonic regions of their screams or squeaks under pain stress in other mice of the same species and show that the ultrasound vocalization of mice under pain affects hyperalgesia in other mice. DNA microarray analysis revealed that the emotional transmission of pain is related to an inflammatory response in the brain thalamus. Our study demonstrated that ultrasonic vocalizations emitted by pain stimuli could result in emotional transmission and cause hyperalgesia.

## Methods

### Materials

Loxoprofen sodium salt dihydrate was purchased from FUJIFILM Wako Pure Chemical Co. (Osaka, Japan). SB225002 was purchased from Sigma-Aldrich (MO, USA). Loxoprofen was dissolved in saline (0.9% sodium chloride) and administered at doses of 300 and 500 mg/kg, intraperitoneally. SB225002 was dissolved in water containing 5% dimethyl sulfoxide (FUJIFILM Wako Pure Chemical Co.), 5% Tween 80 (Tokyo Chemical Industry, Tokyo, Japan), and 90% saline, and administered at doses of 3 and 5 mg/kg, intraperitoneally. The drug doses used in this study were based on those used in previous studies [8].

### Animals

Adult C57BL/6N mice (male, seven weeks old) were obtained from Japan SLC (Shizuoka, Japan) and employed for all behavioral and biochemical studies. The mice were housed at the Institute for Experimental Animals, Tokyo University of Science, under standard conditions (23 ± 1 °C; 12-h light/dark cycle, lights on at 8:00 a.m.) in standard mouse cages with sawdust bedding, and food and water *ad libitum*. We selected a small sample size (n = 6) based on previous research [9], and randomly divided the mice into groups. This study was conducted in accordance with the Fundamental Guidelines for the Proper Conduct of Animal Experiments and Related

Activities in Academic Research Institutions under the jurisdiction of the Ministry of Education, Culture, Sports, Science, and Technology of Japan. All protocols were approved by the Institutional Animal Care and Use Committee of Tokyo University of Science (approval numbers: Y21036 and Y22028).

In this study, humane endpoints were considered, including euthanasia at an appropriate time if the animals experienced intolerable pain or distress. The health and behavior of the animals were monitored daily on weekdays to ensure animal welfare, including checking water bottle usage and food intake. Specifically, humane endpoints were defined as cases where mice exhibit difficulty in feeding or drinking due to stress or pain, or when they lose 25% or more of their body weight. When animals reached the defined endpoints, they were euthanized as quickly as possible. No animals died before meeting the euthanasia criteria.

The duration of each animal experiment was approximately one month each. A total of 177 animals were used, all of which were humanely euthanized. All surgery except for thalamic sample collection was performed under isoflurane anesthesia, with continuous efforts to minimize distress and suffering. When the endpoint was reached, humane euthanasia was carried out without delay. Euthanasia was conducted through isoflurane overdose, a method recognized for being both humane and minimally distressing. Thalamic sample collection was performed by appropriately trained researchers using decapitation without anesthesia. This method was carefully chosen based on evidence that anesthesia can rapidly alter mRNA expression [10]. While minimizing stress and avoiding potential data interference were key considerations, this procedure was conducted strictly under ethical guidelines and with approval from the Institutional Animal Care and Use Committee. All research staff received proper training to ensure the humane handling and care of animals under the guidance of the Institutional Animal Care and Use Committee of Tokyo University of Science and all experiments followed appropriate ethical and technical guidelines.

## Recordings of cries

One of the mice was fixed with a fixation device and the tail was pinched with a 4 cm long artery clip to record its cries (or vocalizations) during pain responses. Cries were recorded using an UltraSoundGate (Avisoft Bioacoustics, Nordbahn, Germany). The ultrasonic regions (over 20 kHz) of the obtained sound sources were used as the sound stress in this experiment.

## von Frey test

Naive mice, different from those used for recording cries, were exposed to sound stress at 80 dB for 4 h (day 0) in a soundproof box. This setup was carefully controlled to deliver consistent sound exposure without additional external stress factors. Sound pressure levels have been reported to influence emotional states in a rat depression model [11]. Tactile thresholds were measured after one, three, and seven days. Inflammatory pain model of the mice was created via injecting 5 µL complete Freund's adjuvant (CFA) into the right hind paw plantar of mice (day 0). The inflammatory pain model mice, which also were different from the ones used for the recordings of cries, were exposed to sound stress at 80 dB for 4 h (day 6) in a soundproof box. Tactile thresholds were measured after 7, 10, 14, 21, and 28 days. Sound stress-induced hyperalgesia and the effects of analgesics were evaluated via the von Frey filaments. The tactile threshold was measured as previously described [12]. A series of von Frey filaments (0.07, 0.16, 0.4, 0.6, 1.0, 1.4, and 2.0 g) were applied (Aesthesio®, DanMic Global, CA, USA). The 50% withdrawal threshold was calculated using the up–down method [13] starting with a 0.6 g filament. If a positive response was observed, the next lower-force filament was applied until a change in response was observed. Four subsequent filaments were then assessed according to the up–down sequence and the 50% paw withdrawal value was calculated using a previously described method [14]. For nociceptive testing, a 0.4 g von Frey filament was placed in the middle of the plantar surface of the ipsilateral hind paw, and a nociceptive test was performed as previously described [12]. Briefly, mice were individually placed in a plastic cage with a wire mesh bottom.

When the mice adapted to the testing environment for 60 min, von Frey filaments were perpendicularly pressed against the mid-plantar surface of the hind paw from below the mesh floor and held for 3–5 s, such that the filament buckled slightly. Paw lifting was recorded as a positive response. Stimulation of the same intensity was applied 10 times to the plantar surface of the ipsilateral hind paw per mouse at 5 s intervals.

### Bioinformatic analysis

The animals were decapitated without anesthesia by appropriately trained researchers to ensure the procedure was performed rapidly and accurately, minimizing stress, and avoiding potential interference of anesthesia with experimental data. The brain regions were then quickly removed and carefully dissected on ice in a modified manner according to Glowinski and Iversen [15], collecting thalamus from mice a day after sound stress exposure for 4 h (day 1) and subjected to microarray analysis. Total RNA isolation and microarray analysis were performed by Hokkaido System Science Co. Ltd. (Hokkaido, Japan). Total RNA was isolated from the thalamus using the RNeasy Lipid Tissue Mini Kit (Qiagen, Netherlands) as per manufacturer's recommendations. RNA quality was checked using a NanoDrop (Thermo Fisher Scientific, MA, USA) and a Bioanalyzer (Agilent Technologies, CA, USA). Labelled cRNA was synthesized from the total RNA using a Low-Input Quick Amp Labeling Kit (Agilent Technologies). The labelled cRNA was hybridized using the SurePrint G3 Mouse 8 × 60 K ver. 2.0 (Design ID: 074809) using a Gene Expression Hybridization Kit (Agilent Technologies). After scanning the slides using an Agilent microarray scanner, a dedicated software was used to quantify the fluorescence signals at each spot. Cy3 and Cy5 signals were compared for each array to identify genes with variable expression.

### Data analysis and statistics

The sample size for each experiment was determined based on our previous studies [9]. Statistical analyses were conducted using the GraphPad Prism 9.0 (GraphPad Software, Inc., CA, USA). All data in the figures are expressed as mean±standard error (SEM). Biochemical and behavioral data were evaluated using a two-tailed *t*-test (two groups) or two-way analysis of variance (ANOVA) *post-hoc* Bonferroni test. The difference was considered statistically significant at a value of $P<0.05$. In each figure, significance denotes $P$ values as follows: $*P<0.05$, $**P<0.01$, $***P<0.001$, and $****P<0.0001$.

## Results

### Sound stress decreased the tactile threshold in mice

We extracted the ultrasonic region (over 20 kHz) of the recorded call and created a psychological stress sound (sound stress) (Fig 1A and 1B), which was applied for 4 h to an individual mouse different from the mouse that uttered the sound (day 0). The results of the von Frey test (Fig 2A) show that the 50% withdrawal threshold decreased significantly on days 1 and 3 of sound stress exposure (Fig 2B; $P<0.0001$ on day 1 and 3 vs day 0); on day 7 of sound exposure, it recovered to the same level as that of the control group. Therefore, we concluded that sound stress led to hyperalgesia.

### Inflammatory cytokines are involved in psychological stress-induced hyperalgesia

To investigate the cause of hyperalgesia induced by sound stress, a microarray analysis was performed using the thalamus, which was removed from mice after 24 h of sound stress exposure for 4 h. We defined upregulated genes as those with gene expression greater than 1.5-fold and downregulated genes as those with gene expression less than 0.67-fold. We identified 56689 genes using microarray analysis. Sound stress exposure resulted in 444 upregulated and 231 downregulated genes compared to those in the control group. A scatter plot of the signal intensities of the dye-corrected gene data is shown in Fig 3A.

The Database for Annotation, Visualization, and Integrated Discovery (DAVID) (https://david.ncifcrf.gov) was used to perform gene ontology (GO) analysis and annotate the function of differentially expressed genes. The upregulated differentially expressed genes (upDEGs) were shown to be significantly associated with annotations in the categories of the

## A

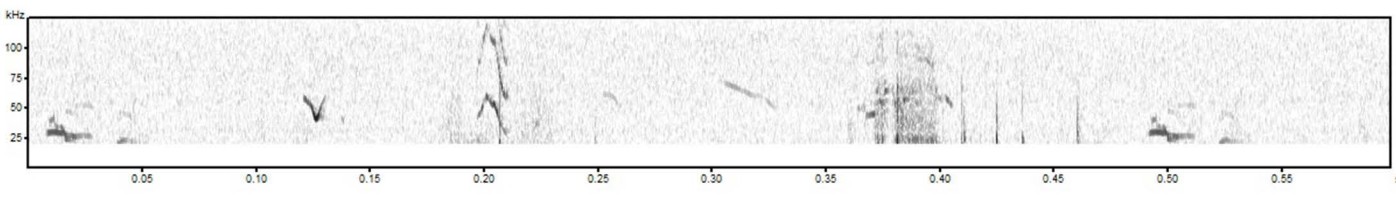

## B

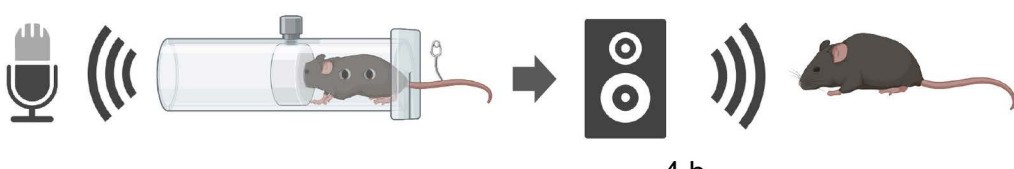

4 h

**Fig 1. Mice vocalizations in ultrasound with pain stimuli.** (A) Example spectrograms of vocalizations emitted by mice exposed to pain stimuli. Ultrasonic regions (over 20 kHz) of the obtained sounds were used as sound stress. (B) Schematic procedure of stress exposure. The mouse was fixed with a fixation device, and its tail was pinched with an artery clip to record its cries (or vocalizations) while in pain. Ultrasonic regions of the obtained sound sources were used as sound stress. Naïve mice, different from those used to record cries, were exposed to sound stress at 80 dB for 4 h in a soundproof box.

regulation of biological processes, cellular components, and molecular functions. On the other hand, none of the down-regulated genes had significantly adjusted *P* values. In biological processes of GO analysis, the upDEGs were associated with the following categories: inflammatory response (e.g., *C-X-C motif chemokine ligand 1 (Cxcl1))*, fold change in sound stress group relative control group (FC), 14.1), response to lipopolysaccharide (e.g., *Prostaglandin-endoperoxide synthase 2 (Ptgs2)*; FC, 17.3), nitric oxide transport (e.g., *Endothelin 1*, FC, 2.5), erythrocyte development (e.g., hemoglobin β adult major chain, FC, 2.3), positive regulation of inflammatory response (e.g., *S100 calcium-binding protein A8*, FC, 12.1), cellular oxidant detoxification (e.g., *Hemoglobin* β *adult major chain*, FC, 2.3) (Fig 3B). GO analysis revealed that many inflammation-related genes were associated with sound stress.

To gain further insight into the function of the upDEGs in the thalamus following sound stress exposure, pathway analysis was conducted, using Kyoto Encyclopedia of Genes and Genomes (KEGG) pathway database (https://www.genome.jp/kegg/pathway.html). Three pathways were identified, indicating regulation by the differentially expressed genes post-sound stress exposure. These genes were primarily linked to the tumor necrosis factor (TNF) signaling pathway (Fig 3C). The 20 genes with the highest rates of signal change are shown in Fig 3D. Of these, five genes (*Ptgs2*, *Cxcl1*, *S100a8*, *Chil3*, and *S100a9*) were classified as inflammation-related clusters (inflammatory and lipopolysaccharide response) in the GO analysis.

Next, to predict protein interactions between the upDEGs, we used the STRING online tool (Search Tool for the Retrieval of Interacting Gene/Proteins) (Fig 4) [16]. The upDEGs formed clusters around genes categorized as inflammatory responses, responses to lipopolysaccharide, and TNF signaling pathways, suggesting that changes in gene expression due to ultrasound exposure were mainly caused by changes in inflammation-related genes.

## Sound stress-induced hyperalgesia can be suppressed with COX-2 or CxCl1 inhibitors

Based on the DNA microarray results, we hypothesized that the COX-2 inhibitor loxoprofen and the CXCL1 inhibitor SB225002 would suppress psychological stress-induced hyperalgesia.

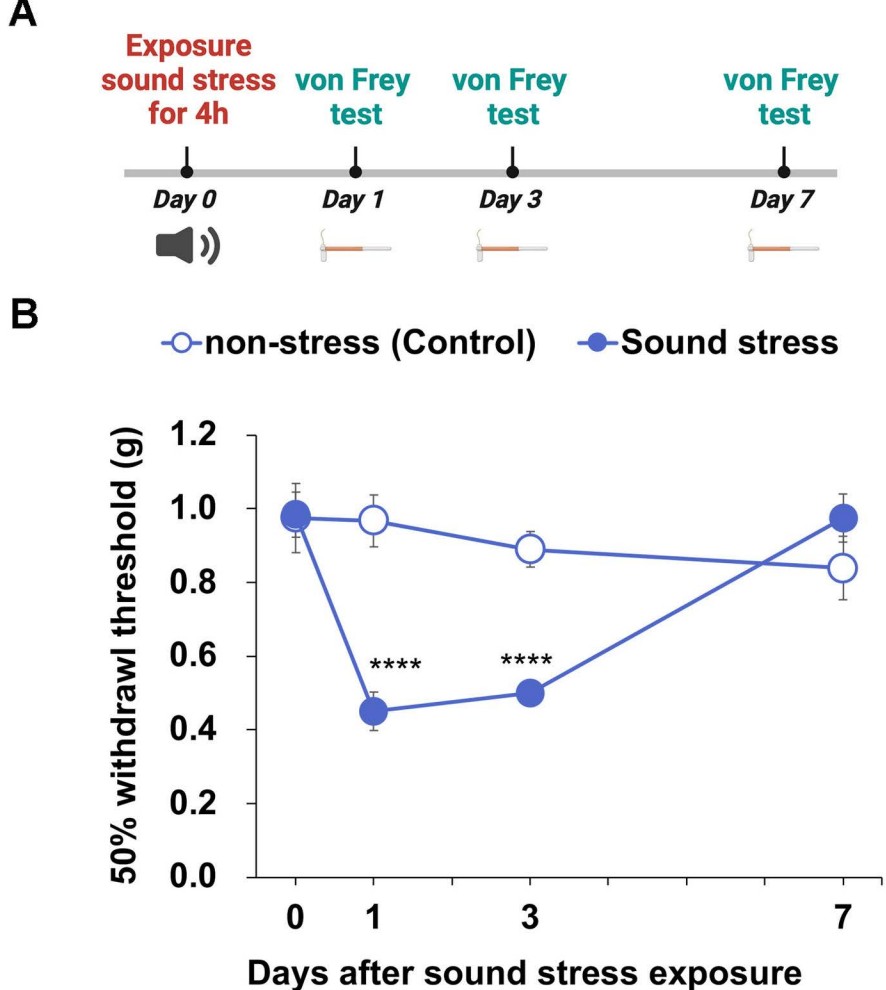

**Fig 2. Sound stress exposure induced hyperalgesia.** (A) Experimental timeline of experiments presented in (B). (B) 50% withdrawal threshold in mice exposed to sound stress. A two-way analysis of variance (ANOVA) showed a significant interaction between the effects of days after sound stress exposure and sound stress ($F_{(3, 40)}$ = 11.27, $P < 0.0001$). ****$P < 0.0001$ vs. day 0 (Dunnett's test). Data are expressed as mean ± SEM (n = 6).

Upon administering them immediately after exposure to psychological sound stress, the results showed that 500 mg/kg loxoprofen at 30 and 60 min post-dosing and 3 mg/kg SB225002 at 60 min post-dosing significantly suppressed pain responses (Fig 5).

**Sound stress prolongs the hyperalgesic period in a mouse model of inflammatory pain**

The mouse model of inflammatory pain, which was exposed to sound stress on day 6 of CFA administration, was assessed for tactile thresholds on days 7–28 using von Frey filaments (Fig 6A). When these mice were exposed to sound stress, the decrease in tactile threshold was prolonged until day 21, indicating a prolonged hyperalgesic effect of CFA (CFA group: $P < 0.0001$ on day 7; CFA + stress group: $P < 0.0001$ on day 7, 10, 14 and 21; $P < 0.0001$; Fig 6B).

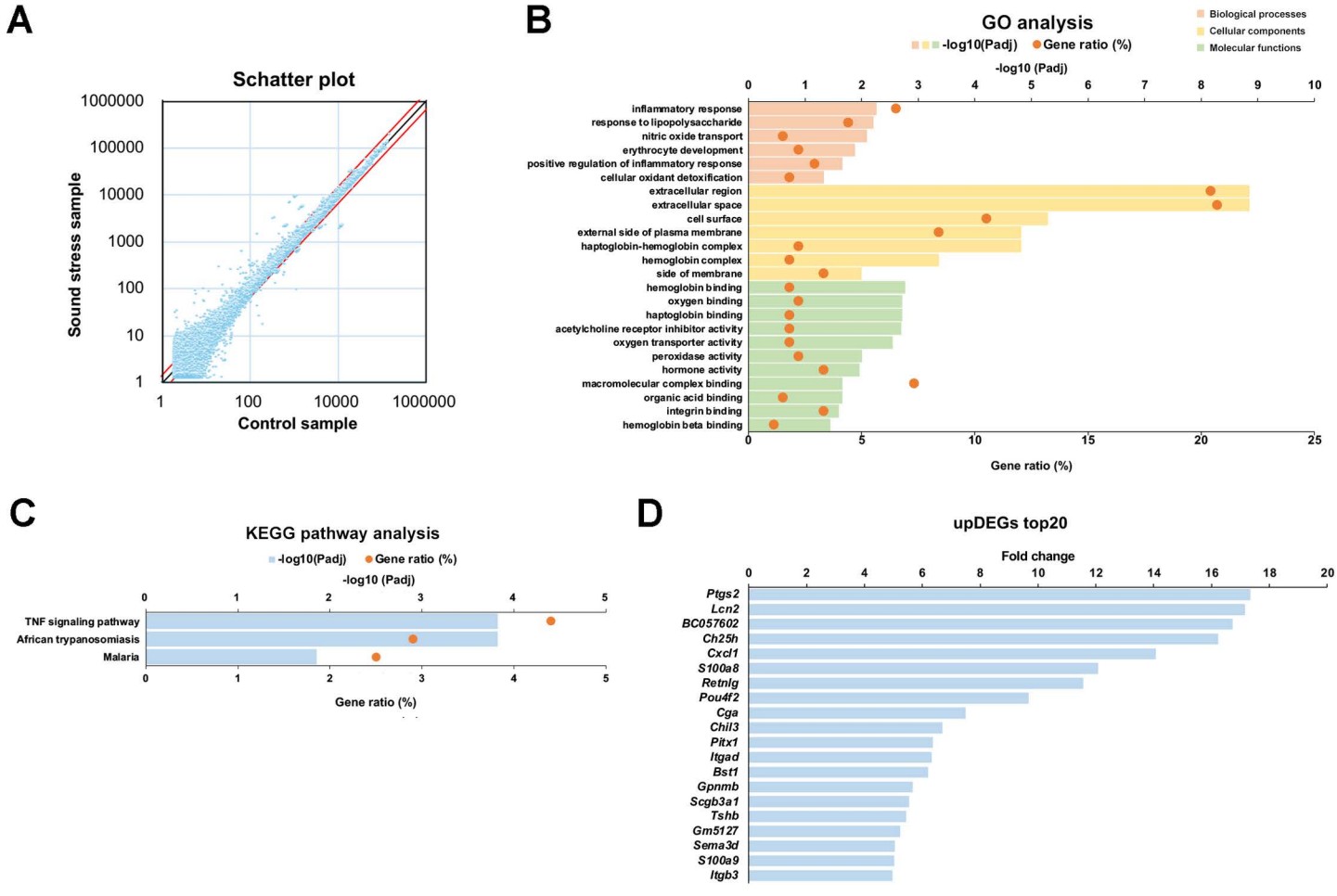

**Fig 3. Sound stress exposure increased inflammatory cytokines in the thalamus.** (A) Scatter plot for visualization gene expression level between control and sound stress samples of the microarray data. The plot represents the normalized values. The red lines indicate the thresholds at which we defined variable expression genes. The upper red line is where gene expression is 1.5-fold and the lower one is where gene expression is 0.67-fold. (B) Gene ontology (GO) annotation for the upregulated differentially expressed genes (upDEGs). The upDEGs were classified into GO categories. (C) Kyoto Encyclopedia of Genes and Genomes (KEGG) pathway analysis for upDEGs. (D) Genes whose expression was more than 1.5-fold because of sound stress exposure (top 20). The *P-adj* values represent the *P*-values after correction for multiple comparisons. *P-adj* < 0.05 is considered significant.

### COX-2 and CxCl1 inhibitors attenuate pain exacerbated by sound stress in a mouse model of inflammatory pain

Next, we assessed the impact of analgesics (loxoprofen and SB225002) on inflammatory pain in CFA exacerbated by sound stress exposure. Administering 500 mg/kg loxoprofen reduced pain responses in the CFA and sound stress-exposed mice groups (*P* < 0.001, 30 and 60 min after injection, Fig 7A). However, the analgesic effect of loxoprofen was attenuated by exposure to sound stress (*P* < 0.001, 60 min after the injection; Fig 7A). Administering 3 mg/kg SB225002 also reduced pain responses in CFA and sound stress-exposed mice (*P* < 0.001, 60 min after injection, Fig 7B).

### Discussion

In this study, we demonstrate for the first time that ultrasonic vocalizations emitted by pain stimuli induced emotional transmission and hyperalgesia in mice exposed to sound stress. These mice exhibit hypersensitivity that arises without injury or direct painful stimulation, but is instead triggered by exposure to sound stress. In the CFA-induced pain mouse model, this

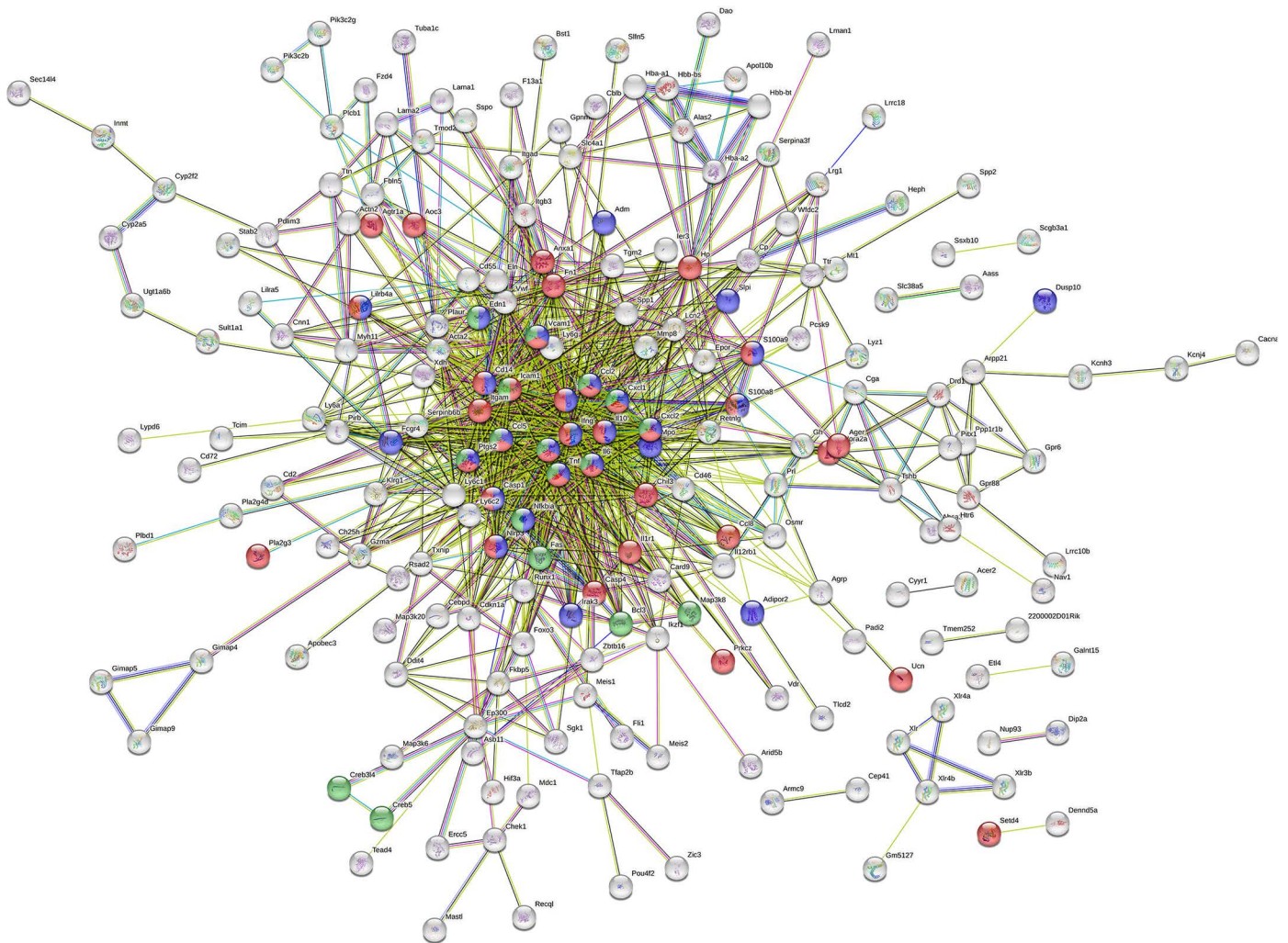

**Fig 4. Functional protein association networks of upregulated genes in the mice exposed to sound stress.** The STRING analysis for a prediction of the connections between the differentially expressed inflammation-related genes. Nodes represent proteins and their colors represent the categories of gene ontology (GO) and Kyoto Encyclopedia of Genes and Genomes (KEGG) pathway analysis: red, gene annotated in inflammatory response in GO analysis; blue, gene annotated in response to lipopolysaccharide in GO analysis; green, gene categorized as tumor necrosis factor signaling pathway in KEGG analysis. The edges represent protein-protein associations and their colors represent the evidence supporting these associations: known interactions from curated database (light blue) and experimentally determined (purple); predicted interactions with gene neighborhood (green), gene fusions (red), and gene co-occurrence (blue); genes with text mining evidence (yellow green); co-expression genes (black); protein homology (light purple).

phenomenon was observed in delayed recovery from pain, that is, a prolonged period of pain. Furthermore, sound stress exacerbates inflammatory pain and reduces analgesic efficiency.

Several studies have investigated the social transfer of pain; olfactory signals facilitate the transfer of hyperalgesia in observer mice [4]; and mice that have been housed together for several weeks with other mice of the same species having peripheral nerve damage exhibit heightened responses in the acetic acid-induced writing test [17]. In the present study, the mice were exposed only to the sound emitted by other mice with pain stimuli in the absence of another individual mouse showing a pain response, thus eliminating sensory modalities other than hearing, such as sight and smell, and demonstrating that the pain transfer phenomenon could occur solely through auditory input.

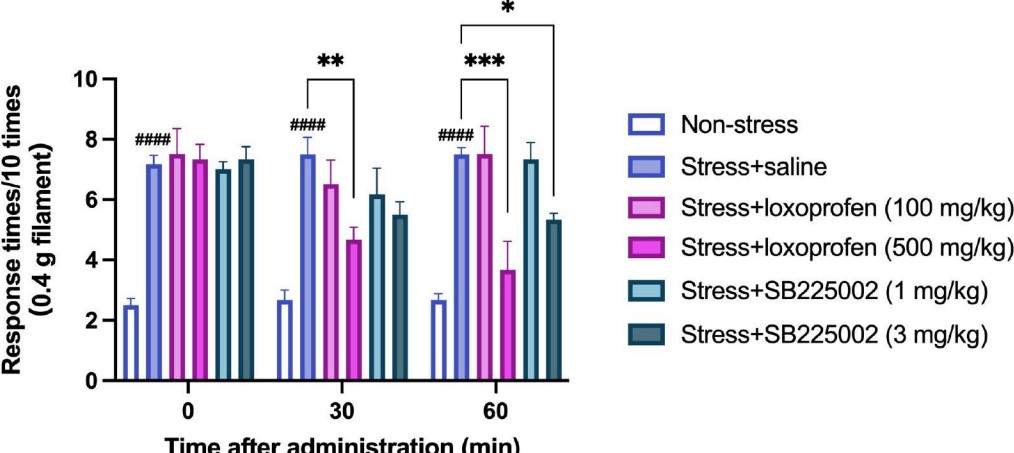

**Fig 5. Loxoprofen and SB225002 administration decreased sound stress exposure-induced hyperalgesia.** The nociceptive test assessed the effects of analgesics after sound exposure. Mice exposed to sound stress and treated with saline (Stress+saine group) showed significantly increased response times to the filament compared to the control (non-stress) group (0–30 min after administration, ####$P < 0.0001$; Bonferroni test). Mice treated with loxoprofen (500 mg/kg) or SB225002 (3 mg/kg) showed significantly decreased response times to the filament compared to Stress+saline group (Stress+loxoprofen (500 mg/kg), 30 min after administration, **$P < 0.01$, 60 min after administration, ***$P < 0.001$; Stress+SB225002 (3 mg/kg), 60 min after administration, *$P < 0.05$). 2-way ANOVA *post-hoc* Bonferroni test. Data are expressed as mean±SEM (n=6). Non-stress, intact mouse; Stress+saline, mice exposed to sound stress and treated with saline; Stress+loxoprofen, mice exposed to sound stress and treated with loxoprofen; Stress+SB225002, mice exposed to sound stress and treated with SB225002.

Future experiments comparing specific frequency tones with pain stimulus tones are needed to determine whether hyperalgesia occurs due to ultrasonic vocalizations conveying pain-related information, making the mice aware of danger, or due to specific ultrasonic frequencies directly inducing hyperalgesia. In depression-like model rats, exposure to 50 kHz ultrasound vocalizations associated with positive emotions, as well as artificial 50 kHz ultrasound, induced antidepressant-like effects [18]. Therefore, the possibility that ultrasound of specific band frequencies may be involved in hyperalgesia induction cannot be ruled out.

Ultrasounds at frequencies above 20 kHz are inaudible to humans; however, ultrasound exposure has been reported to activate the reward system in the human brain (hypersonic effect) [19]. Additionally, a study by Yamauchi et al. reported that inaudible ultrasound at a frequency of 100 kHz affected emotional states in rats [11]. Therefore, the emotional changes induced by ultrasound exposure may not necessarily depend on the audibility of the ultrasounds. In our study, the frequency of the exposed sounds was above 20 kHz. The audible range of mice is broadly extensive; from 1 kHz to about 100 kHz [20]. Thus, we consider that the sound may be partially audible to mice; however, we have not yet confirmed the detailed audibility of the sounds used in our study. Further studies are needed to investigate this aspect.

Microarray analysis of the thalamus of mice exposed to sound stress revealed increased expression of several genes associated with inflammation. The brain thalamus is the site where pain impulses travel along afferent nociceptive fibers from the periphery when nociceptors are stimulated; further neurotransmission occurs to the cerebral cortex, where pain is perceived [21]. Sound stress increased the expression of inflammation-related genes in the brain. Our findings indicate that the heightened brain inflammation triggered by sound stress exposure contributes to hyperalgesia and worsens inflammatory pain.

Loxoprofen, a COX-2 inhibitor, exhibits analgesic effects in carrageenan-injected rats at 3–30 mg/kg orally. In the air pouch model rats with inflammation, the ED50 values for inhibition of air pouch PGE2 and gastric mucosa PGE2 were 2.0 and 2.1 mg/kg, respectively [22]. In this study, we demonstrated the analgesic effects of high doses of loxoprofen (500 mg/kg) with no notable adverse effects observed. Loxoprofen (0.3–30 mg/kg) is known to be distributed throughout the brain,

**A**

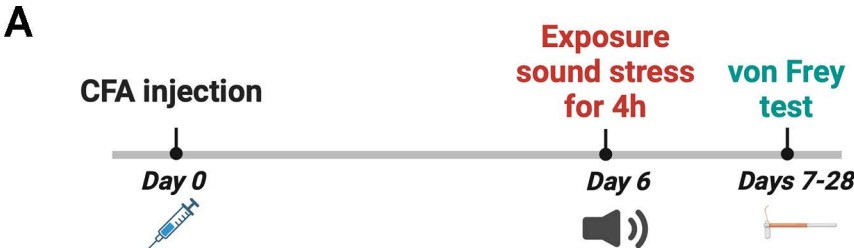

**B**

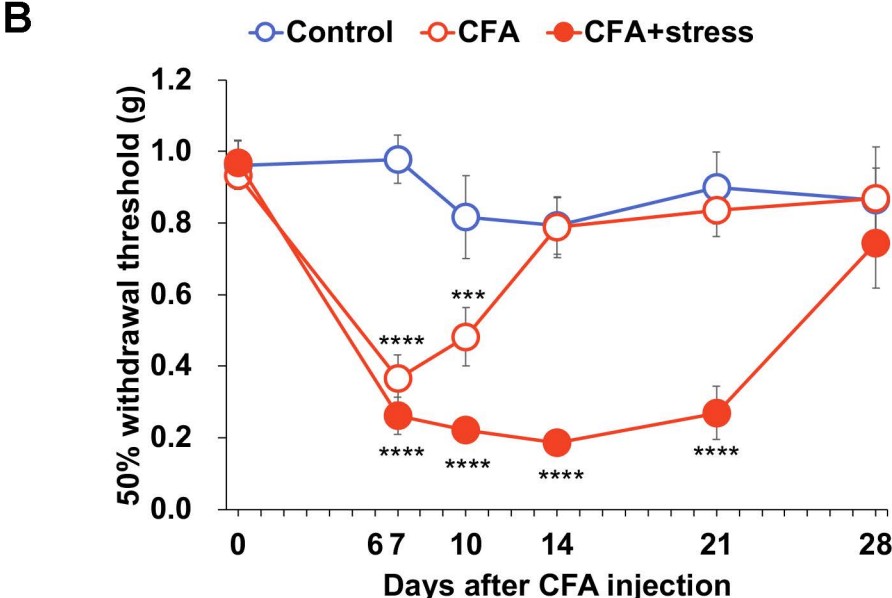

**Fig 6. Sound stress exposure prolonged the period of decreased withdrawal threshold after complete Freund's adjuvant (CFA) injection.**
(A) Timeline of experiments shown in (B). (B) 50% withdrawal threshold in CFA-treated mice exposed to sound stress. They showed a significant decrease in 50% withdrawal threshold on day 7 after CFA injection, and CFA-treated mice exposed to sound stress showed a significant decrease in 50% withdrawal threshold during days 7–21 after CFA injection (CFA, on day 7, ****$P < 0.001$ vs day 0, on day 10, ***$P < 0.001$ vs day0; CFA+stress, day 7, 10, 14, and 21, ****$P < 0.0001$ vs day 0, Dunnett's test). Data are expressed as mean±SEM (n=6). Control, control mice; CFA, CFA-treated mice; CFA+stress, CFA-treated mice with sound stress exposure.

and to inhibit brain PGE2 [23]. Therefore, our findings suggest that high-dose loxoprofen inhibits not only peripheral PGE2, but also brain PGE2.

Stress exposure induces brain inflammation and worsens inflammation symptoms. Restraint and sound stress exposure activates the toll-like receptor (TLR)-4 pathway in the prefrontal cortex via activating microglia and mast cells, causing brain inflammation and blood-brain barrier disruption [24]. Repeated sound stress in rats increased inflammatory pain via altering the bradykinin-induced hyperalgesia mechanism [25]. In the KEGG analysis, the TLR-4 pathway was not significantly related to upDEGs; however, in GO analysis, genes were categorized in response to lipopolysaccharide, stimulating TLR-4 receptors, and activated by microglia and mast cells [24]. Therefore, the emotional transmission observed in our study may also involve microglia and mast cell activation.

CXCL1 is a CXC chemokine and activates the CXC motif chemokine receptor 2 (CXCR2) in humans [26]. SB225002 is a CXCR2 antagonist that reduces neutrophil migration to the brain in mice with neuroinflammation [27]. In this study, the expression of several chemokinetic genes was upregulated (S1 Table). The secreted proinflammatory cytokines (such

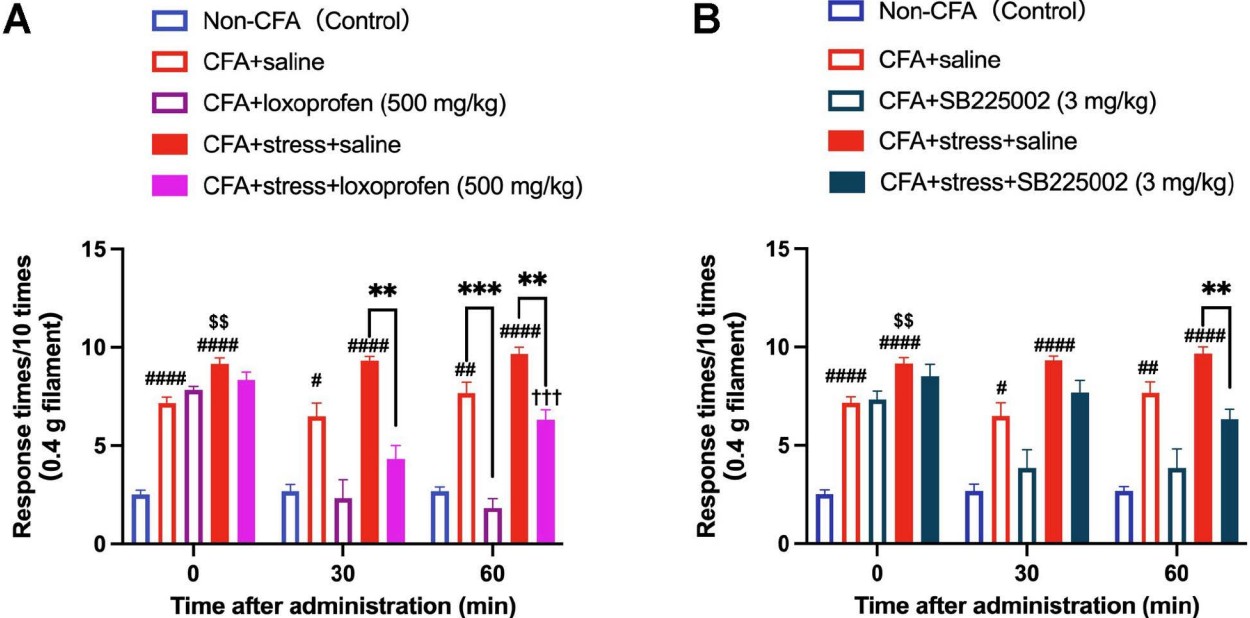

**Fig 7. Loxoprofen or SB225002 administration attenuated inflammatory pain in complete Freund's adjuvant (CFA)-injected mice exposed to sound stress.** The nociceptive test assessed the effects of (A) loxoprofen and (B) SB225002 after sound exposure in CFA-treated mice. (A, B) The CFA-injected mice (CFA+saline) and CFA-injected mice with sound stress exposure (CFA+stress+loxoprofen) showed significantly increased response times to filament compared to the control (non-CFA) group (CFA+saline, 0 min, ####$P < 0.0001$; 30 min, #$P < 0.05$; 60 min, ##$P < 0.01$; CFA+stress+saline, 0, 30, and 60 min, ####$P < 0.0001$). CFA-injected mice with sound stress exposure showed significantly increased response times to filament compared CFA+saline group (0 min, $$$P < 0.01$). **(A)** Loxoprofen administration (500 mg/kg) significantly decreased the response times to filament in CFA+saline group 60 min after administration (***$P < 0.001$), and in CFA+stress+saline group (30 and 60 min after administration, **$P < 0.01$). The effects of loxoprofen were significantly reduced by sound stress exposure (60 min, †††$P < 0.001$). **(B)** SB225002 administration (3 mg/kg) significantly decreased the response times to filament in CFA+stress+saline group (60 min, $P = 0.0037$). Data are expressed as mean ± SEM (n = 6).

as TNF-α, interleukin [IL]-6, IL-12, IL-18) that were responsible for continuing inflammation were not detected; however, chemokines (such as Cxcl1, Ccl5, Ccl8) that were responsible for recruiting immune cells, which could be measured as M1 microglia markers [28], were detected. The expression of M1 microglial markers, proinflammatory cytokines, and chemokines were elevated, leading to systemic inflammation [29]. The hyperalgesia observed in this study was alleviated by SB225002, a CXCR2 antagonist, suggesting that social transfer observed may be associated with neuroinflammatory responses in the thalamus, potentially involving microglial activity. However, further experiments are required to confirm a causal relationship, particularly regarding the balance between M1 and M2 microglia. Nonetheless, the results indicate that CXCR2 antagonists may represent a promising new therapeutic strategy for mitigating hyperalgesia induced by ultrasound exposure.

Pain can be classified into nociceptive, neuropathic, and nociplastic pain. According the International Association for the Study of Pain, nociplastic pain is "pain that arises from altered nociception despite no clear evidence of actual or threatened tissue damage causing the activation of peripheral nociceptors or evidence for disease or lesion of the somatosensory system causing the pain" [30]. We clarified that the pain induced by ultrasonic exposure cannot be strictly defined as nociplastic pain, given the increased inflammation gene expression in the thalamus of exposed mice. However, our results highlight a novel aspect of pain modulation through ultrasonic vocalizations, suggesting that ultrasound exposure may influence pain perception via neuroinflammatory mechanisms. Further investigations, including assessments of neuronal hyperexcitability and glial activation, will be essential to fully elucidate the underlying pathways. Understanding these

mechanisms may pave the way for new therapeutic strategies targeting ultrasound-induced pain modulation, ultimately contributing to advances in pain management.

## Supporting information

**S1 Table. Upregulated genes in the thalamus of the mouse exposed sound stress (fold change > 1.5).**
(XLSX)

## Acknowledgments

The authors would like to acknowledge the continuous support of all members involved in the collaborative research on ultrasound between FUJIMIC Inc. and the Tokyo University of Science. We would like to thank Editage (www.editage.jp) for English writing editing. Some figures were created using BioRender.com.

## Author contributions

**Conceptualization:** Kazumi Yoshizawa.

**Data curation:** Satoka Kasai.

**Formal analysis:** Satoka Kasai.

**Funding acquisition:** Masashi Suzuki, Satoru Miyazaki.

**Investigation:** Satoka Kasai, Saki Ukai, Junpei Kuroda, Kazumi Yoshizawa.

**Methodology:** Saki Ukai, Junpei Kuroda, Kazumi Yoshizawa.

**Project administration:** Satoka Kasai, Kazumi Yoshizawa.

**Supervision:** Satoka Kasai, Tsugumi Yamauchi, Daisuke Yamada, Akiyoshi Saitoh, Satoshi Iriyama, Masashi Suzuki, Yoshio Nakano, Satoru Miyazaki, Kazumi Yoshizawa.

**Visualization:** Satoka Kasai.

**Writing – original draft:** Satoka Kasai.

**Writing – review & editing:** Satoka Kasai, Kazuki Arita.

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
