## [Decision Letter · Decision Letter 0]

Pain-stimulated ultrasound vocalizations and their impact on pain response in mice

Dear Dr. Kasai,

**Comments from the Academic Editor:**

The methods lack details that should be incorporated. For instance, it is unclear if they are providing sound stress, so what decibel range are the authors using to provide that stress, and for how long?

Authors should report how they minimize suffering, which anesthetic agents they use, and how they euthanize mice. How did they measure difficulty in feeding or drinking, and why would the animals kept alive for up to 2 hours if they figured the mouse was in pain?

We look forward to receiving your revised manuscript.

Kind regards,

Santosh Kumar Mishra

Academic Editor

PLOS ONE

“I have read the journal's policy and the authors of this manuscript have the following competing interests: [This work was supported by a grant from FUJIMIC Inc. (Tokyo, Japan) and AMED-CREST under Grant Number JP23gm1510008s0102. MS is an employee of FUJIMIC Inc.]”

Reviewers' comments:

Reviewer's Responses to Questions

**Comments to the Author**

1. Is the manuscript technically sound, and do the data support the conclusions?

Reviewer #1: Yes

Reviewer #2: Yes

2. Has the statistical analysis been performed appropriately and rigorously?

Reviewer #1: Yes

Reviewer #2: Yes

3. Have the authors made all data underlying the findings in their manuscript fully available?

Reviewer #1: Yes

Reviewer #2: Yes

4. Is the manuscript presented in an intelligible fashion and written in standard English?

Reviewer #1: Yes

Reviewer #2: Yes

Reviewer #1: -The paper of Kasai S et al. is relevant and interesting, of interest of academic community. The text is well written, and the design well conducted. Some minor points are presented below.

1. The paper of Kosek et al. (ref. 2), mentioned in the introduction should be checked again, and the definition of nociplastic pain completely revised and rewrite;

2. Please, provide details on the experimental procedures applied for collecting samples from the thalamus, with the respective reference;

3. In the final paragraph of the discussion section, authors claim that this study provides a model of ultrasound vocalizations-induced nociplastic pain. The Reviewer is not convinced that such a description is adequate. Although in the present model, there is no substantial damage to peripheral tissues, the authors did not conduct additional experiments to assess whether thalamic and/or spinal neurons exhibit hyperexcitability. Immunofluorescence using antibodies for neuronal markers in the spinal cord and/or thalamus (e.g., anti-NeuN) must confirm this condition.

In this sense, it would be worth mentioning that inflammatory pain (due to the elevated detection of some inflammatory markers) induced by ultrasonic vocalizations was observed. And not claim that it is nociplastic pain. This could be a description of one of the limitations of the study that the authors may mention, which could be answered in future studies.

Additionally, as molecular experiments targeting glia have not been conducted either, the Reviewer suggests caution in the line of reasoning for the participation of microglia. Perhaps it is hypothesized that this indicates the participation of glial cells in the model, but that additional experiments would be necessary for such confirmation.

Reviewer #2: Ideally, an experiment using randomized (non-pain related) ultrasonic vocalizations would have been included; however, I find the authors' self-critique of this missing data to be acceptable. Overall, this is an interesting study.

**Do you want your identity to be public for this peer review?** For information about this choice, including consent withdrawal, please see our Privacy Policy

Reviewer #1: No

Reviewer #2: No

---

## [Author Response · Author response to Decision Letter 1]

25 Feb 2025

Responses to Reviewer #1:

Thank you very much for reviewing of our manuscript.

Your comments are highly insightful and enable us to improve the quality of our manuscript. Our responses to the respective comments are described in the following columns. Incorporating these responses, we have revised our manuscript, thus, we would like to resubmit it for publication in PLOS ONE.

1. Definition of Nociplastic Pain:

We have carefully reviewed the paper by Kosek et al. and accordance with reviewer’s comment, we judged that we cannot exactly define the pain induced by ultrasound exposure as nociplastic pain. Therefore, we deleted the sentences which were related with nociplastic pain, and revised in “Introduction” (page 4, lines 39 – 44) as follows:

“Introduction” (page 4, lines 39 – 44) Pain is an important physiological phenomenon that can be a major stressor, sometimes to the point of death, and can occur even in the absence of actual tissue damage [1]. Many reports show that psychological stress enhances pain [2, 3]. In 2016, Smith et. al. demonstrated that “bystander” mice housed in the same environment as mice undergoing inflammatory pain exhibited similar hyperalgesia [4]. This emotional transmission is thought to occur because of various information (sight, hearing, smell, etc.) obtained from others; however, their mechanisms are unclear.

2. Experimental Procedures for Thalamus Sample Collection:

Detailed information on the procedures used to collect thalamic samples has been added. The animals were decapitated without anesthesia by appropriately trained researchers to ensure the procedure was performed rapidly and accurately, minimizing stress, and avoiding potential interference of anesthesia with experimental data. The brain regions were then carefully dissected based on anatomical landmarks. The revised text includes a reference to the protocol used in the “Methods” (page 9, lines 128 – 131) as follows:

“Methods” (page 9, lines 128 – 131) The animals were decapitated without anesthesia by appropriately trained researchers to ensure the procedure was performed rapidly and accurately, minimizing stress, and avoiding potential interference of anesthesia with experimental data. The brain regions were then quickly removed and carefully dissected on ice in a modified manner according to Glowinski and Iversen [15], collecting thalamus from mice a day after sound stress exposure for 4 h (day 1) and subjected to microarray analysis.

3. Model Description and Limitations:

We acknowledge the reviewer's concerns regarding the characterization of this model as one of nociplastic pain. As we did not assess neuronal hyperexcitability in the hypothalamus and/or spinal cord, we recognize that this study alone does not provide sufficient evidence to definitively classify this model as nociplastic pain. In the revised discussion, we emphasize that our primary focus is on pain-related ultrasonic vocalizations and acknowledge that additional experiments—such as assessments of neuronal hyperexcitability and glial activation—are necessary to fully validate the relevance of this model to nociplastic pain.

Furthermore, we agree with the reviewer’s comment that the pain induced by ultrasonic exposure cannot be strictly defined as nociplastic pain, given the observed increase in inflammation-related gene expression in the thalamus of exposed mice. Therefore, our results should be interpreted as revealing a novel pain mechanism induced by ultrasound exposure rather than as definitive proof of nociplastic pain. In accordance with the reviewer's comment, we have revised the manuscript in both “Introducion” and “Discussion” sections to reflect these points.

“Introduction” (page 4, lines 54 – 56) Our study demonstrated that ultrasonic vocalizations emitted by pain stimuli could result in emotional transmission and cause hyperalgesia.

“Discussion” (page 21, lines 360 – page 22, lines 370) Pain can be classified into nociceptive, neuropathic, and nociplastic pain. According the International Association for the Study of Pain, nociplastic pain is “pain that arises from altered nociception despite no clear evidence of actual or threatened tissue damage causing the activation of peripheral nociceptors or evidence for disease or lesion of the somatosensory system causing the pain” [30]. We clarified that the pain induced by ultrasonic exposure cannot be strictly defined as nociplastic pain, given the increased inflammation gene expression in the thalamus of exposed mice. However, our results highlight a novel aspect of pain modulation through ultrasonic vocalizations, suggesting that ultrasound exposure may influence pain perception via neuroinflammatory mechanisms. Further investigations, including assessments of neuronal hyperexcitability and glial activation, will be essential to fully elucidate the underlying pathways. Understanding these mechanisms may pave the way for new therapeutic strategies targeting ultrasound-induced pain modulation, ultimately contributing to advances in pain management.

4. Glial Cell Involvement Hypothesis:

We agree with your comment. Following the reviewer’s comment, we have adopted a more cautious approach when discussing the involvement of microglia. The revised discussion now frames this as a hypothesis that warrants further investigation rather than a definitive conclusion in “Discussion” (page 21, lines 353 – 359) as follows:

“Discussion” (page 21, lines 353 – 359) The hyperalgesia observed in this study was alleviated by SB225002, a CXCR2 antagonist, suggesting that social transfer observed may be associated with neuroinflammatory responses in the thalamus, potentially involving microglial activity. However, further experiments are required to confirm a causal relationship, particularly regarding the balance between M1 and M2 microglia. Nonetheless, the results indicate that CXCR2 antagonists may represent a promising new therapeutic strategy for mitigating hyperalgesia induced by ultrasound exposure.

Responses to Reviewer #2:

We appreciate the reviewer’s acknowledgment of our self-critique regarding the absence of randomized, non-pain-related ultrasonic vocalizations as a control. While we agree that this is a limitation, we believe our current discussion adequately addresses this point and highlights it as an area for future research.

---

## [Decision Letter · Decision Letter 1]

Pain-stimulated ultrasound vocalizations and their impact on pain response in mice

PONE-D-24-35208R1

Dear Dr. Kasai,

We’re pleased to inform you that your manuscript has been judged scientifically suitable for publication and will be formally accepted for publication once it meets all outstanding technical requirements.

Kind regards,

Santosh Kumar Mishra

Academic Editor

PLOS ONE

Additional Editor Comments (optional):

Reviewers' comments:

Reviewer's Responses to Questions

**Comments to the Author**

Reviewer #1: All comments have been addressed

2. Is the manuscript technically sound, and do the data support the conclusions?

Reviewer #1: Yes

3. Has the statistical analysis been performed appropriately and rigorously?

Reviewer #1: Yes

4. Have the authors made all data underlying the findings in their manuscript fully available?

Reviewer #1: Yes

5. Is the manuscript presented in an intelligible fashion and written in standard English?

Reviewer #1: Yes

Reviewer #1: Authors adequately answered the question and concerns regarding the paper. The paper is now suitable for publication.

**Do you want your identity to be public for this peer review?** For information about this choice, including consent withdrawal, please see our Privacy Policy

Reviewer #1: No

---

## [Editor Report · Acceptance letter]

PONE-D-24-35208R1

PLOS ONE

Dear Dr. Kasai,

I'm pleased to inform you that your manuscript has been deemed suitable for publication in PLOS ONE. Congratulations! Your manuscript is now being handed over to our production team.

Kind regards,

on behalf of

Dr. Santosh Kumar Mishra

Academic Editor

PLOS ONE